# The Influence of Craniometric Variation on the Position of Mandibular Foramen: A Cadaveric Cross-Sectional Study

**DOI:** 10.3390/medicina60121925

**Published:** 2024-11-23

**Authors:** Hadi Darawsheh, Ali Alsaegh, Vladimir Nikolenko, Saida Bakieva, Irina Smilyk, Andrew Panin, Artur Kheygetyan, Vasiliy Troitskiy, Dmitry Leonov, Yuriy Vasil’ev

**Affiliations:** 1Department of Operative Surgery and Topographic Anatomy, First Moscow State Medical University (Sechenov University), 119435 Moscow, Russia; bakieva_s_i@student.sechenov.ru (S.B.); troitskiy_v_i@staff.sechenov.ru (V.T.); leonov_d_s@staff.sechenov.ru (D.L.); vasilev_yu_l@staff.sechenov.ru (Y.V.); 2Medical Institute, Patrice Lumumba Peoples’ Friendship University, 117198 Moscow, Russia; ali0alsaegh@gmail.com; 3Department of Anatomy and Histology, First Moscow State Medical University (Sechenov University), 119435 Moscow, Russia; nikolenko_v_n@staff.sechenov.ru (V.N.); 4Institute of Anatomy, 121205 Skolkovo, Russia; smilyk_i@mail.ru; 5Russian University of Medicine, 119991 Moscow, Russia; andreypanin@yandex.ru; 6Dentistry Department No.1, Rostov State Medical University, 344022 Rostov-on-Don, Russia; artur5953@yandex.ru

**Keywords:** mandible, mandibular foramen, mandibular nerve, mandibular canal, nerve block, face, skull, inferior alveolar nerve, anthropometry, education

## Abstract

*Background and Objectives*: the mandibular foramen is an essential anatomic landmark in performing various dental and surgical procedures, including inferior alveolar nerve block (IANB). However, its position may vary based on the individual morpho-functional features of the skull and face. This study aims to conduct a personalized assessment of the location of the mandibular foramen in various shapes of skulls, faces, and mandibles. *Materials and Methods*: this anatomic morphometric cross-sectional study was performed using one hundred and six (*n* = 106) certified human cadaver heads of both sexes. The cranial index (CI) and Izard’s facial index (FI) were calculated, the linear anatomic parameters of the skull and mandible were measured, the location of the mandibular foramen was identified, and the shapes of the skulls and mandibles were determined. Quantitative statistical data were obtained based on the location of the mandibular foramen, considering different shapes of skulls and faces. *Results*: there is a significant correlation between the location of the mandibular foramen, the high lengthy index (HLI) of the mandible, and the longitudinal latitude index (LLI) of the mandibular process. *Conclusions*: personalized assessment of the mandibular foramen based on a comprehensive analysis of craniometric characteristics can contribute to preventing unwanted dental and surgical complications, such as inferior alveolar nerve damage.

## 1. Introduction

The shapes of heads and mandibles have individual variations that depend on various factors, such as gender, age, and ethnicity [1,2,3,4]. The geometric morphometric characteristics of the skull and their variations are needed to carry out a craniometric analysis, which comprises measuring the dimensions of the skull [5,6].

The standard craniometric landmarks (points) include the glabella (the point between the two superciliary arches), metopion (midline landmark where there is elevation above the chord from nasion to bregma), bregma (the midline landmark where the coronal and sagittal sutures meet), lambda (midline landmark where the lambdoid sutures and sagittal suture meet), inion (the tip of the external occipital protuberance), basion (the midline point of the anterior margin of the foramen magnum), gonion (most posterior inferior point on the mandibular angle), occiput (the occipital bone; often the most prominent posterior point of the occiput), vertex (the highest point of the cranium), and nasion (the midline landmark just superior to the nasal root) [6,7,8,9]. All the above-mentioned landmarks are essential for measuring the size of the skull.

The cranial (cephalic) index (CI) can be calculated using the following formula: CI = (the transverse diameter × 100)/the longitudinal diameter. The transverse diameter is the distance between the parietal eminences (between the euryons). The longitudinal diameter is the distance from the glabella to the external occipital protuberance. The CI is a valid and reliable mathematical index for identifying the shapes of the skull: dolichocrania or dolichocephaly (CI = 77), mesocrania or mesocephaly (CI = 77–80), and brachycrania or brachycephaly (CI = 80 or above) [10,11].

The main craniometric parameters of the face are two: (1) the facial width, which can be measured using the maximum distance between the most lateral points on the zygomatic arches (zy–zy), and (2) the morphological facial height, which can be measured as the vertical distance from the nasion to the gnathion. The FI is a valid and reliable mathematical index for identifying the following three types of faces: euryprosopic (FI: 84–85), mesoprosopic (FI: 85–90), and leptoprosopic (FI: 90 or above). The FI can be calculated from the mathematical formula: FI = the facial length × 100/the facial width [10].

The facial angle (the intersection of the line passing through the nasion and prosion) and the Frankfort horizontal (a plane that intersects both porions and the left orbitale) are reliable and valid in identifying the shapes of the facial skull: prognathism (up to 800); mesognathism (80–850); orthognathism (850 or above) [12,13].

Studies suggested various classifications to determine mandibular shapes, the mandibular length, the condylar and angular width, and the distance between the mental foramina. The study of Brandsburg B.B. (1931), based on the value of the latitude longitudinal index (LLI), classified mandibular shapes into brachymandibular and dolicholmandibular [14]. In 2012, Gladilin Y.A. published a study that classified mandibular shapes into oval, trapezoid, and triangular [15].

In the early 1970s, Kuznetsova L.V. proposed a simplified classification that identified mandibular shapes: short and wide, and narrow and long. In addition, Kuznetsova revealed a correlation between the mandibular shape and the shape of the upper part of the facial skull; the leptoprosopic facial skeleton shape is more often associated with a small mandibular angular width, and the euryprosopic facial skeleton shape is more often associated with a significant mandible width [16]. However, the study of Tvardovskaya M.V. (1972) proposed a classification to distinguish between rectangular, v-shaped, and sled-shaped mandibles; this classification emphasizes the possibilities of different morphological changes throughout life under the influence of several external factors [17].

With growing interest in translational dental medicine, the study of Smirnov V.G. et al. (2014) suggested that, for clinical dentistry, the most appropriate classification of mandibular shapes is: (1) wide and short; (2) long and narrow [18]. Besides classifying the mandibular shapes, various published studies have proposed and validated metric systems for the comprehensive and evidence-based morphometric evaluation of the mandible [19,20]. One of the high-quality, valid systems comprises three indices: the high lengthy index (HLI), the longitudinal latitude index (LLI), and latitudinal altitude index (LAI) [21,22,23].

The mandibular foramen is located above the center of the medial surface of the mandibular ramus [24]. It transmits the inferior alveolar nerve, artery, and vein. Its terminal branch emerges from the mental foramen. The two mandibular foramina are interconnected by the mandibular canal [25,26]. These data play a role in the carrying out of a safe inferior alveolar nerve block: they help to avoid positive aspiration during injections and nerve injury. Previously published studies have proven that the anatomic location of the mandibular foramen varies; it is located 10–25 mm from the mandibular anterior edge, 9–20 mm from the mandibular posterior edge, 17–29 mm from the mandibular notch, and 15–35 mm from the mandibular angle [27,28].

Localizing the mandibular foramen has theoretical and clinical benefits; it expands the knowledge on the individual variability of the shape of the skull and improves the safety of dentistry. The mandibular foramen is an important anatomic landmark for the safe performance of dental and surgical procedures, such as inferior alveolar nerve block (IANB), the placement of dental implants, and mandibular osteotomy [29,30,31,32,33,34,35].

Clinicians and translational researchers must identify the individual features of the location of both the mandibular foramen and the mandibular canal [36]. A lack of interdisciplinary awareness of the variations in these landmarks can lead to clinical complications, for example increasing the risk of damage to the lower alveolar nerve during dental and surgical procedures [37]. Failure to localize the mandibular foramen increases the risk of injury to the maxillary artery, temporomandibular joint, lateral pterygoid muscle, and sphenomandibular ligament.

Avoiding errors is a top-priority issue in dentistry and medicine. There is no doubt that the need to improve patient safety strongly justifies the need for a comprehensive personalized assessment of the location of the mandibular foramen based on a comprehensive craniometric analysis. This study aims to identify if there is a correlation between the location of the mandibular foramen and the shapes of skulls, faces, and mandibles.

## 2. Materials and Methods

### 2.1. Study Design and Ethics

This cadaveric geometric morphometric descriptive cross-sectional study aimed to assess the location of the mandibular foramen in different shapes of skulls, faces, and mandibles. The Research Ethics Committee of I.M. Sechenov First Moscow State Medical University (Sechenov University) approved this study (26 January 2023; 02–23). We conducted the present study according to the ethical principles of the World Medical Association Declaration of Helsinki (1964).

### 2.2. Legal Aspects

The transfer of an unclaimed body, organs, and tissues of a deceased individual for use in scientific purposes is regulated by Article 68 of the Russian Federal Law of 21 November 2011 No. 323-FZ on the fundamentals of health protection of citizens in the Russian Federation. According to this article, if the body is not claimed after the death of an individual due to the absence of his spouse, close relatives, other relatives, legal representatives, or other individuals who have assumed responsibility for the burial, the maximum period of use of the unclaimed body, organs, and tissues of a deceased individual cannot exceed 10 years. After this period, the unclaimed body is subject to burial by the legislation of the Russian Federation on burial and funeral services, which is regulated by Federal Law No. 8-FZ of 12 January 1996 (as amended on 6 April 2024) on burial and funeral services (as amended, supplemented, and entered into force on 1 September 2024).

### 2.3. Population/Sample

We conducted this study at the Department of Operative Surgery and Topographic Anatomy, N.V. Sklifosovskiy Institute of Clinical Medicine, I.M. Sechenov First Moscow State Medical University (Sechenov University). We investigated one hundred and six certified human cadavers (*n* = 106) of both sexes (males and females) from the collection of the Skolkovo Institute of Anatomy, Skolkovo, Russia. The mean age of the investigated cadavers was 72.67 years (SD = 11.47). The inclusion criteria were: age over 18 years old and no fractures in, tumors on, or traumatic injuries of the oral and maxillofacial region. We excluded all cadavers that did not meet the inclusion criteria.

### 2.4. Anatomic Measurements and Dissection

We measured Izard’s facial index (FI) and the cranial (cephalic) index (CI) [10]. We performed cadaveric dissection to extract the mandibles, and the soft tissues were fully removed. We used a digital caliper (Taiwan) to take all the measures (the measurement range = 0–15 cm/0–6′ and the error rate = 0.01 mm/0.0005′.)

The following mandibular parameters were measured (Figure 1): (1) the location of the mandibular foramen; (2) the distance from the mandibular angle to the mandibular lingula; (3) the distance from the mandibular foramen to the mandibular lingula and the coronoid process; (4) the distance from the mandibular foramen to the mandibular lingula and the condylar process; (5) the distance from the mandibular foramen to the mandibular lingula and the anterior edge of the mandible; (6) the distance from the mandibular foramen to the mandibular lingula and the mandibular notch; (7) the thickness of the mandibular process through the mandibular lingula.

The investigated cadavers, based on their mandibular shape, were divided into three morphometric indices (Figure 2):High lengthy index (HLI) of the mandible;Longitudinal latitude index (LLI) of the mandibular body;Latitudinal altitude index (LAI) of the mandibular process [10].

The values of four parameters were used to calculate the morphometric indices:The projection length from the mandibular angles is the distance from the pogonion (the most anterior point of the chin protrusion in the median section) to the middle of the line between both gonions (points on the outer edge of the mandible at its intersection with the bisector of the angle formed by the tangents to the lower edge of the body and the rear edge of a mandibular process);The mandibular angular width (the distance between the gonions);The mandibular process height (the distance from the gonion to the upper point of the condyle, parallel to the posterior edge of the process);The smallest width of the process (the smallest distance between the front and rear edges of the process).

Calculation of the morphometric indices:1.The HLI of the mandible can be estimated from the following formula: HLI = the height of the mandible process/the projection length from the mandibular angles × 100. The HLI is reliable in identifying three types of mandibular shapes: dolichogenia, mediagenia, and brachygenia (Figure 3);

2.The LLI of the mandibular body can be estimated from the flowing formula: LLI = the ratio of the projection length from the corners/the angular width × 100. This index allows systemizing the mandible only according to the shape of its body, not considering the parameters of the mandible process. The LLI is reliable in identifying three types of mandibular shapes: (A) leptogenia, (B) mesogenia, and (C) eurygenia (Figure 4);

3.The LAI of the mandibular process can be estimated from the flowing formula: LAI = the ratio of the smallest process width/its height × 100. This index allows systemizing the mandible based on the shape of the processes, without considering the parameters of the body. The LAI is reliable in identifying three types of mandibular shapes: hypsigenia, orthogenia, and platygenia (Figure 5) [21,23].

### 2.5. Statistical Analysis

The statistical data obtained in this study were tabulated using Microsoft Office Excel 365 (Microsoft Corporation, Redmond, WA, USA) and then analyzed using R4.1.2 (RStudio Desktop 1.1.463) statistical software package (Posit, Boston, MA, USA). The sample size was 84 (population proportion is 50%, population size is 106, confidence level 95%, margin of error 5%).

Q–Q graphs and the Shapiro–Wilk test were implemented to detect deviations from the normal distribution. Continuous data were presented as M ± SD (M is the mean value, and SD is the standard deviation) or Me (IQR) (Me is the median, and IQR is the interquartile range). Qualitative data were presented as P ± σp (P is the percentage, and σp is the standard deviation). The groups were compared using nonparametric and parametric criteria. Correction of multiple comparisons was performed using the Benjamini-Hochberg procedure. Relationships were studied using linear regression and correlation analysis (the Pearson product-moment correlation coefficient (r) and Spearman’s rank correlation coefficient were calculated, with the determination of the 95% confidence interval (CI) and *p*-value). As part of the regression analysis, when necessary, interaction terms were introduced to the model. Differences were considered statistically significant at *p* < 0.05.

## 3. Results

Based on the data obtained using the CI, the following head shapes were identified: dolichocephaly (49.06%), mesocephaly (28.30%), and brachycephaly (22.64%). Based on the data obtained using the FI, the following face shapes were identified: broad (wide) facial type (30.19%), medium facial type (15.09%), and narrow facial type (54.72%).

Based on the data obtained using the morphometric indices (HLI, LLI and LAI), the following results for mandibular shapes were identified (Table 1):The HLI on the right: mediagenia (60.38%), dolichogenia (35.84%), and brachygenia (3.77%);The HLI on the left: mediagenia (58.49%), dolichogenia (32.07%), and brachygenia (9.43%);The LLI: eurygenia (0%), mesogenia (13.21%), and leptogenia (86.79%);The LHI on the right: platygenia (45.28), orthogenia (49.06%), and hypsigenia (5.66%);The LAI on the left: platygenia (37.74), orthogenia (50.94%), and hypsigenia (11.32%).

The results presented in Table 2 show the locations of mandibular foramina obtained using various anatomic landmarks.

The results presented in Table 3 show all the correlations between the location of the mandibular foramen and the CI.

The results presented in Table 3 show that most correlations were found using the HLI and LAI, and no statistically significant correlations with the parameters of interest were found using the CI.

The results presented in Table 4 show the location of the mandibular foramen in different shapes of the mandible. It was found that, for the dolichogenia (on the right), the maximum values were observed in four out of six parameters, and a similar result was observed regarding the measurements on the left side. For the platygenia (on the right side), the maximum values were observed for the distances from the mandibular angle to the mandibular lingula and from the mandibular lingula to the coronoid process. For hypsigenia, in comparison with other mandibular shapes, the maximum values were correlated with the distance from the mandibular lingula to the mandibular anterior edge and the width of the mandibular process.

## 4. Discussion

Failure in localizing the mandibular foramen is a primary risk factor that contributes to decreasing the success rate of IANB and increasing the complications of dental procedures [38,39]. In a published retrospective study (T.M. You et al., 2015), the authors investigated 693 mandibular third molar dental extractions followed by performing IANB. Based on the values of the condyle–coronoid ratio, they suggested a classification method for mandibular shapes: orthognathic, prognathic, and retrognathic. The statistical results of You et al. showed that the frequency of unsuccessful IANB was higher for the retrognathic mandibular shape than for the orthognathic mandibular shape, and compared with the prognathic mandibular shape, the results showed no significant differences. The failure rate of IANB was the highest at a condyle–coronoid ratio <0.8, which corresponded to a severe retrognathic mandibular shape. The authors concluded that the effectiveness of local anesthesia during dental procedures depends on the morphological characteristics of the mandible [40].

The height of the mandibular process depends mainly on the bone growth of the mandibular angle, and this causes an expansion in the space between the mandibular foramen and the mandibular angle [41]. In the current study, it is possible to compare the present data with the results published by Gladilin Y.A. et al. (2020), who found a positive correlation between head circumference and mandibular size; different mandibular sizes were also not correlated [15].

Published studies have proven that tooth loss decreases the space between the mandibular fossa and the posterior and inferior borders of the ramus and causes ramus remodeling with a reduction in the bony structures of the ramus in the mandibular fossa depression [21,42,43]. With factors like aging and tooth loss, the angle of the mandibular process increases and results in the displacement of the mandibular foramen in two directions (posteriorly and downwards) [27].

According to a published literature review (D.Y. Choi et al., 2021), the mandibular lingula is an important anatomic landmark, and dentists must consider its location while performing IANB. Proper evaluation of the mandibular ligula improves the effectiveness and reduces the failure rate of IANB. The authors of the study identified several anatomic features of the mandibular ligula based on the mandibular shapes; on a prognathic mandible, the lingula is located posteriorly and higher than on non-prognathic mandibles; most mandibular ligulas are above the occlusal plane, but are sometimes on or below the occlusal plane. The triangular ligula was usually located somewhat higher and more posteriorly than in other forms [44]. In the literature, there is a lack of evidence showing which anatomic landmarks are the most optimal for accurate localization of the mandibular foramen.

A morphometric study (K. Sandhya et al., 2015) analyzed the location of the mandible foramen relative to six bony landmarks on the ramus. The distance from the mandibular foramen to all landmarks, except for the condyle, was symmetrical on both sides. The authors concluded that the condyle and the mandibular internal oblique ridge are new landmarks that can localize the mandibular foramen [38]. In the present study, the results showed no significant differences between the morphometric parameters used to determine the position of the mandibular foramen on both sides of the mandible. Also, our results revealed that the most reliable parameters in localizing the mandibular foramen were the HLI and LAI indices, which determine the shape of the mandible.

Studies have proven that, besides the individual anatomic variation in the mandibular foramen, some mandibles present accessory mandibular foramina, which can also cause difficulties in performing dental and maxillofacial surgical procedures. Of interest in this regard is a published morphometric study (R. Shalini et al., 2016) in which the authors investigated two hundred and four Indian dry human mandibles, and the results showed that an accessory mandibular foramen was present in 32.36% of all the included mandibles [27].

Studies have shown that the shape of the skull affects the shape and length of the mandibular canal and the location of the inferior alveolar nerve [45,46]. A published observational retrospective study (R. Dos Santos Oliveira et al., 2018) concluded that the morphology of the mandibular canal and its variations present significant associations with different facial types [47]. However, the results of the present study revealed no significant correlations between the location of the mandibular foramen and the facial index.

Integrating the results of the present morphometric research paper into medical and dental education would have significant benefits, and underestimating the role of anatomic variations in education is a catastrophic issue that results in a reported and unreported burden of unsafe care [48,49]. An example is a published study of malpractice insurance claims from four North American insurance corporations that reported that 25 percent of claims were associated with morphometric variations, and the mandibular foramen is no exception [48,50]. Another example is a recent systematic review (Nzenwa IC et al., 2023) that aimed to explore the literature on anatomical variation in medical education and showed that there is a significant gap regarding the topic (only eight studies were eligible) [48].

The present study is the first assessment of the position of the mandibular foramen that considered anthropometric variations (the shapes of skulls, faces, and mandibles); previous studies in the English literature did not implement similar materials and methods. This study has strengths, including the substantial sample size (106 human cadaver heads) and the detailed morphometric analysis. This study has limitations, including sample diversity (the study did not explore how variables like ethnicity or gender could influence mandibular foramen location) and the exclusion of dysgnathia and skeletal anomalies (skeletal anomalies were not a focus).

Since this study relied on cadaveric measurements without imaging techniques, there may be limits in applying these findings to live patient procedures. However, this study is the first step in answering the research question. There is a need for similar future studies that might support developing high-quality, objective, reliable, and transparent systemic literature reviews and meta-analyses.

## 5. Conclusions

The significant anatomical variability in mandibular shapes can affect clinical dental and surgical practice. The personalized assessment of mandibular foramen based on the evidence-based analysis of patients’ craniometric characteristics can prevent damaging mandibular neurovascular bundles and reduce the frequency of dental and surgical complications.

It is essential to enhance patient safety through education, and an up-to-date integration of translational research on the anatomical variations of the mandibular foramen can strengthen dental education, develop clinical reasoning, and improve the outcomes of related procedures, such as IANB. The present study attempts to establish needed resources for anatomical variation education and can benefit medical educators in curriculum development and shaping educational programs.

## Figures and Tables

**Figure 1 medicina-60-01925-f001:**
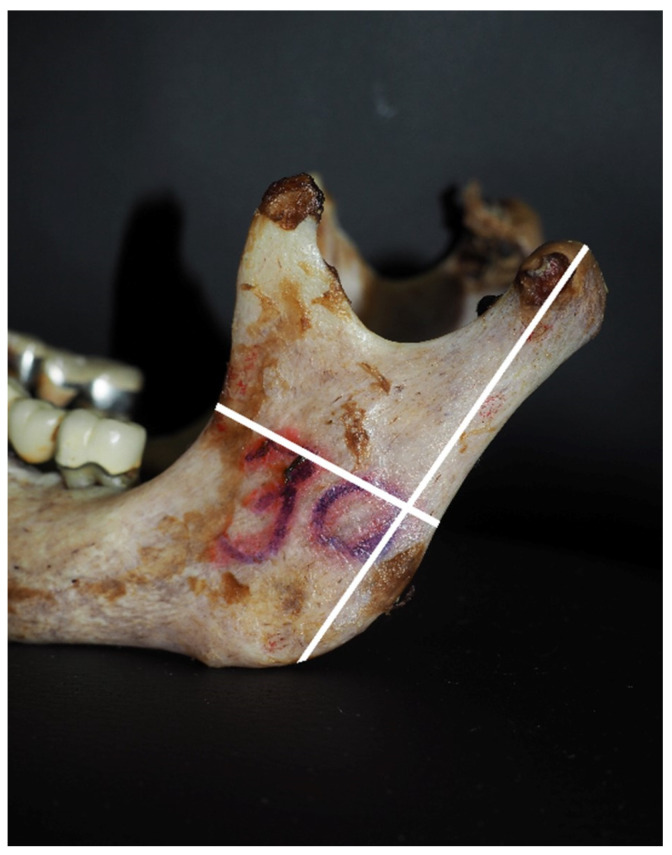
Measured anatomic parameters on the extracted mandibles.

**Figure 2 medicina-60-01925-f002:**
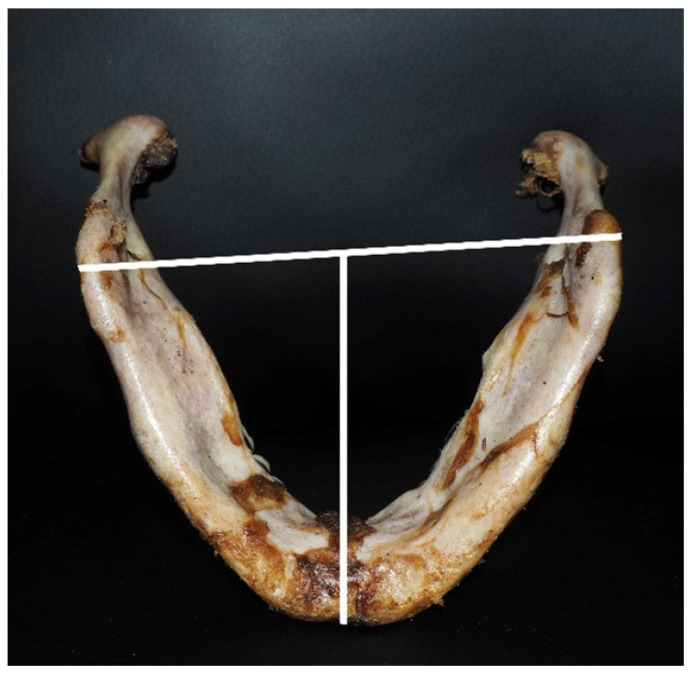
The method used to measure the mandibular shapes.

**Figure 3 medicina-60-01925-f003:**
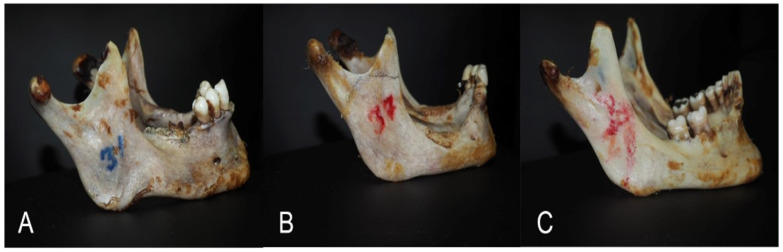
The mandibular shapes based on the HLI of the mandible: (**A**) dolichogenia, (**B**) mediagenia, and (**C**) brachygenia.

**Figure 4 medicina-60-01925-f004:**
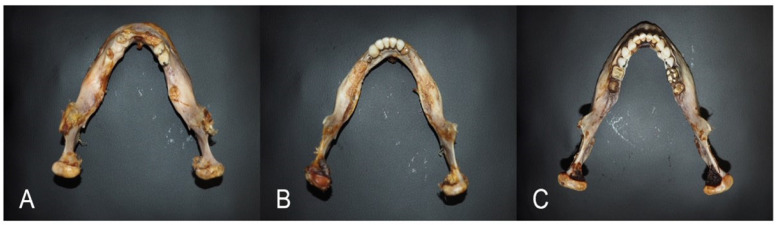
The mandibular shapes based on the LLI of the mandibular body: (**A**) leptogenia, (**B**) mesogenia, and (**C**) eurygenia.

**Figure 5 medicina-60-01925-f005:**
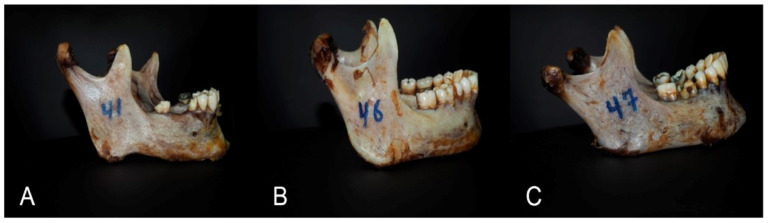
The mandibular shapes based on the LAI of the mandibular process: (**A**) hypsigenia, (**B**) orthogenia, and (**C**) platygenia.

**Table 1 medicina-60-01925-t001:** Comparative characteristics of morphometric parameters for determining mandibular shapes (*p* < 0.05).

Parameter	$\bar{x}$ ± SD		Me (IQR)	Minimum	Maximum
Projection length from the mandibular angles	6.64 ± 0.63		6.7 (6.3–7.0)	5.1	7.7
Mandibular angular width	9.78 ± 0.75		9.9 (9.4–10.3)	8.2	11.4
Height of mandibular process	Right	6.54 ± 0.57	6.5 (6.1–6.9)	5.5	7.8
	Left	6.49 ± 0.57	6.4 (6.1–6.9)	5.4	7.8
Smallest width of the mandibular process	Right	3.02 ± 0.42	3.0 (2.8–3.3)	2.1	3.9
	Left	3.0 ± 0.43	3.0 (2.7–3.4)	2.2	3.7

**Table 2 medicina-60-01925-t002:** Descriptive statistics for the parameters of the locations of mandibular foramina (*p* < 0.05).

Parameter		$\bar{x}$ ± SD	Me (IQR)	Minimum	Maximum
Distance from the mandibular angle to the mandibular lingula	Right	3.09 ± 0.38	3.1 (2.81–3.40)	2.11	3.89
Distance from the mandibular angle to the mandibular lingula	Left	3.04 ± 0.42	3.07 (2.09–3.94)	2.09	3.94
Distance from the mandibular lingula to the coronoid process	Right	3.57 ± 0.39	3.58 (2.48–4.44)	2.48	4.44
Distance from the mandibular lingula to the coronoid process	Left	3.67 ± 0.41	3.65 (2.60–4.56)	2.60	4.56
Distance from the mandibular lingula to the condylar process	Right	3.84 ± 0.32	3.81 (3.56–4.14)	3.13	4.53
Distance from the mandibular lingula to the condylar process	Left	3.88 ± 0.37	3.88 (3.56–4.11)	2.85	4.57
Distance from the mandibular lingula to the anterior edge of the mandible	Right	1.63 ± 0.23	1.63 (1.52–1.87)	1.3	2.27
Distance from the mandibular lingula to the anterior edge of the mandible	Left	1.69 ± 0.21	1.68 (1.56–1.80)	1.27	2.32
Thickness of the mandibular process (through the mandibular lingula)	Right	3.28 ± 0.31	3.02 (2.76–3.20)	2.41	3.88
Thickness of the mandibular process (through the mandibular lingula)	Left	3.05 ± 0.30	3.07 (2.87–3.21)	2.51	4.05
Distance from the mandibular lingula to the mandibular notch	Right	1.89 ± 0.33	1.89 (1.71–2.12)	1.13	2.51
Distance from the mandibular lingula to the mandibular notch	Left	1.91 ± 0.37	1.96 (1.67–2.19)	0.97	2.52

**Table 3 medicina-60-01925-t003:** The correlations between the location of the mandibular foramen and the CI.

Parameter	CI	FI	HLI	LLI	LAI
Right side					
Distance from the mandibular angle to the mandibular lingula	-	-	ρ = +0.543 *p* = <0.0001	-	ρ = −0.237 *p* = 0.0143
Distance from the mandibular lingula to the coronoid process	-	-	ρ = +0.319 *p* = 0.0009	-	-
Distance from the mandibular lingula to the condylar process	-	-	ρ = +0.339 *p* = 0.0004	-	-
Distance from the mandibular lingula to the anterior edge of the mandible	-	-	-	-	ρ = +0.366 *p* = 0.0001
Thickness of the mandibular process	-	-	-	ρ = +0.193 *p* = 0.04779	ρ = +0.448 *p* < 0.0001
Distance from the mandibular lingula to the mandibular notch	-	-	-	-	ρ = −0.192 *p* = 0.0481
Left side					
Distance from the mandibular angle to the mandibular lingula	-	-	ρ = +0.454 *p* < 0.0001	-	ρ = −0.229 *p* = 0.0184
Distance from the mandibular lingula to the coronoid process	-	ρ = −0.201 *p* = 0.03861	ρ = +0.239 *p* = 0.01376	-	ρ = −0.271 *p* = 0.005
Distance from the mandibular lingula to the condylar process	-	-	ρ = +0.363 *p* = 0.000132	-	-
Distance from the mandibular lingula to the anterior edge of the mandible	-	-	-	-	ρ = +0.273 *p* =0.0045
Thickness of the mandibular process	-	ρ = +0.215 *p* = 0.0266	-	ρ = +0.213 *p* = 0.02833	ρ = +0.385 *p* < 0.0001
Distance from the mandibular lingula to the mandibular notch	-	-	ρ = +0.342 *p* = 0.0003	-	ρ = −0.440 *p* < 0.0001

The table indicates statistically significant correlations. ρ is the Spearman’s rank correlation coefficient, + is the direct relationship, and - is the inverse relationship.

**Table 4 medicina-60-01925-t004:** Location of the mandibular foramen in different mandibular shapes (*p* < 0.05).

HLI of the Mandible	Dolichogenia		Mediagenia		Brachygenia	
Side	Right	Left	Right	Left	Right	Left
Distance from the mandibular angle to the mandibular lingula	3.33 ± 0.36	3.28 ± 0.41	2.99 ± 0.30	2.97 ± 0.34	2.41 ± 0.15	2.65 ± 0.45
Distance from the mandibular lingula to the coronoid process	3.78 ± 0.37	3.80 ± 0.43	3.46 ± 0.37	3.63 ± 0.40	3.35 ± 0.24	3.52 ± 0.35
Distance from the mandibular lingula to the condylar process	3.96 ± 0.30	4.02 ± 0.34	3.77 ± 0.33	3.78 ± 0.32	3.64 ± 0.09	3.70 ± 0.58
Distance from the mandibular lingula to the anterior edge of the mandible	1.77 ± 0.25	1.78 ± 0.24	1.62 ± 0.19	1.62 ± 0.16	1.92 ± 0.05	1.83 ± 0.13
Thickness of the mandibular process (through the mandibular lingula)	3.07 ± 0.37	3.11 ± 0.40	2.98 ± 0.27	3.02 ± 0.24	3.27 ± 0.17	3.12 ± 0.21
Distance from the mandibular lingula to the mandibular notch	1.96 ± 0.33	2.04 ± 0.42	1.86 ± 0.32	1.87 ± 0.32	1.57 ± 0.36	1.71 ± 0.22
LLI of the Mandibular Body	Leptogenia		Mesogenia		Eurygenia	
Side	Right	Left	Right	Left	Right	Left
Distance from the mandibular angle to the mandibular lingula	3.09 ± 0.38	3.06 ± 0.41	3.07 ± 0.41	2.90 ± 0.47	±	±
Distance from the mandibular lingula to the coronoid process	3.59 ± 0.37	3.69 ± 0.38	3.47 ± 0.53	3.59 ± 0.60	±	±
Distance from the mandibular lingula to the condylar process	3.85 ± 0.32	3.85 ± 0.37	3.78 ± 0.35	3.82 ± 0.40	±	±
Distance from the mandibular lingula to the anterior edge of the mandible	1.68 ± 0.22	1.69 ± 0.19	1.71 ± 0.28	1.69 ± 0.30	±	±
Thickness of the mandibular process (through the mandibular lingula)	3.04 ± 0.28	3.06 ± 0.27	2.97 ± 0.51	3.06 ± 0.47	±	±
Distance from the mandibular lingula to the mandibular notch	1.90 ± 0.32	1.92 ± 0.33	1.83 ± 0.40	1.83 ± 0.51	±	±
LAI	Hypsigenia		Orthogenia		Platygenia	
Side	Right	Left	Right	Left	Right	Left
Distance from the mandibular angle to the mandibular lingula	2.66 ± 0.40	2.72 ± 0.44	3.04 ± 0.32	3.05 ± 0.31	3.20 ± 0.40	3.11 ± 0.50
Distance from the mandibular lingula to the coronoid process	3.42 ± 0.21	3.51 ± 0.31	3.60 ± 0.42	3.57 ± 0.42	3.56 ± 0.38	3.86 ± 0.36
Distance from the mandibular lingula to the condylar process	3.83 ± 0.31	3.79 ± 0.57	3.76 ± 0.36	3.78 ± 0.35	3.92 ± 0.28	3.95 ± 0.31
Distance from the mandibular lingula to the anterior edge of the mandible	1.99 ± 0.11	1.90 ± 0.20	1.68 ± 0.21	1.68 ± 0.21	1.65 ± 0.24	1.64 ± 0.15
Thickness of the mandibular process (through the mandibular lingula)	3.47 ± 0.34	3.27 ± 0.41	3.06 ± 0.27	3.11 ± 0.25	2.93 ± 0.30	2.91 ± 0.28
Distance from the mandibular lingula to the mandibular notch	1.91 ± 0.33	1.91 ± 0.31	1.81 ± 0.39	1.82 ± 0.50	1.66 ± 0.27	1.73 ± 0.31

## Data Availability

The original contributions presented in the study are included in the article, further inquiries can be directed to the corresponding author/s.

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
