# Peer review of "The Influence of Craniometric Variation on the Position of Mandibular Foramen: A Cadaveric Cross-Sectional Study"

_medicina, 2024, doi:10.3390/medicina60121925_

Round 1
Reviewer 1 Report
Comments and Suggestions for Authors
INTRODUCTION
Line 94 please correct English.
Overall introduction is well-described although I suggest adding more clinical implications associated with mandibular foramen localization. Also, I suggest adding anatomical structures localized in the area of mandibular foramen.
Please add aims and objectives of this study.
MATERIALS AND METHODS
Materials and methods are well-described and include all the studied characteristics. I suggest adding a table with the values of parameters.
RESULTS
Results are well-represented.
DISCUSSION
Discussion includes limitations of this study and also analyzes pertinent literature.
CONCLUSION
I suggest improving conclusions answering the question “what do we learn in this paper?”; I also suggest including future directions.
Author Response
Dear reviewer!
We appreciate your time and attention to our manuscript. Thanks to your important comments we were able to significantly improve the text and make it more understandable to readers.
1. Line 94 was fixed.
2. Overall introduction is well-described although I suggest adding more clinical implications associated with mandibular foramen localization. Also, I suggest adding anatomical structures localized in the area of mandibular foramen
We added this information at the lines 86-90
3. Please add aims and objectives of this study.
Added at the lines 111-112
4.I suggest improving conclusions answering the question “what do we learn in this paper?”; I also suggest including future directions.
Please find changes at the lines 366-371
Reviewer 2 Report
Comments and Suggestions for Authors
Can Personalized Assessment of the Mandibular Foramen in Different Shapes of Skulls, Faces, and Mandibles Minimize the Risk of Inferior Alveolar Nerve Injury?
Reviewer Report
The title does not fully reflect the study. Please provide a fluent and clear title.
The abstract has been presented in a sufficiently informative manner.
Craniometric landmarks, cranial (cephalic) index, craniometric parameters of the face, and facial angle have been presented more like the Materials and Methods section rather than providing background relevant to the research questions in the Introduction. Please provide more information on their significance.
The rationale and clinical contribution of this study should be emphasized further. Why was this study needed?
The aim of the study should be restated at the end of the Introduction section.
In the Materials and Methods section, how was the sample size calculated? The power analysis of the study should be specified.
The criteria for including cadavers in the study should be clearly specified.
How reliability was ensured in repeated measurements should be specified.
In the Results section, tables and the overall appearance of the text should be adapted to the journal template.
Table 3 should be revised for clarity, as its current format makes it difficult to understand.
The p-values should be displayed in the necessary tables for the statistical comparison results.
The Discussion section is insufficient and should be expanded further to establish a closer connection to the literature.
The limitations and strengths of the study should be emphasized.
Citations need to be revised to comply with the journal’s guidelines.
This conclusion could be stated without the need for this study. Please provide a more concrete conclusion/recommendation that aligns with the study's objectives based on its results.
Revise the References according to the journal’s guidelines. (For example, reference number 16)
Comments on the Quality of English LanguageModerate-level English language edits are necessary.
Author Response
Dear reviewer!
We appreciate your time and attention to our manuscript. Thanks to your important comments we were able to significantly improve the text and make it more understandable to readers.
- The title does not fully reflect the study. Please provide a fluent and clear title.
The title was changed into "The Influence of Craniometric Variation on the Position of Mandibular Foramen: A Cross-Sectional Cadaveric Study"
2. Craniometric landmarks, cranial (cephalic) index, craniometric parameters of the face, and facial angle have been presented more like the Materials and Methods section rather than providing background relevant to the research questions in the Introduction. Please provide more information on their significance.
We wrote that all the above-mentioned landmarks are essential for measuring the size of the skull.
3. The rationale and clinical contribution of this study should be emphasized further. Why was this study needed?
Please find our comments at the lines 86-90
4. The aim of the study should be restated at the end of the Introduction section.
It was replaced at the lines 111-112
5. In the Materials and Methods section, how was the sample size calculated? The power analysis of the study should be specified.
Please find this at the lines 205-206
6. The criteria for including cadavers in the study should be clearly specified.
We added that info at the lines 137-139
7. How reliability was ensured in repeated measurements should be specified.
All the measurements were taken using a digital caliper (line 143) 8. In the Results section, tables and the overall appearance of the text should be adapted to the journal template. We fixed that data 9. Table 3 should be revised for clarity, as its current format makes it difficult to understand. We think that current version shows data better 10. The p-values should be displayed in the necessary tables for the statistical comparison results. This information was written in Materials and methods (line 217-218) but we added it to the tables descriptions 11. The Discussion section is insufficient and should be expanded further to establish a closer connection to the literature. We added required information at the lines 340-360 12. The limitations and strengths of the study should be emphasized. Please find it at the lines 137-139 13. Citations need to be revised to comply with the journal’s guidelines. Thank you, we fixed it. 14. This conclusion could be stated without the need for this study. Please provide a more concrete conclusion/recommendation that aligns with the study's objectives based on its results. It was clarified at the lines 366-371Round 2
Reviewer 2 Report
Comments and Suggestions for Authors
The sample size is not clear. Please clarify it more explicitly in order to eliminate the ambiguity. To do this, refer to a previous similar study.
In the revision comment, when it was requested to specify how reliability is measured in repeated measurements, it was not meant to refer to the name and accuracy of the measuring device. Reliability in repeated measurements means the following: you take an initial measurement, and to test whether this measurement is reliable, you perform a remeasurement on 20-25% of the total sample randomly selected after a period of time (for example, after 2-4 weeks). Then, you demonstrate the reliability of the measurements by presenting the intra-class correlation coefficients (ICCs) for the two measurements. Please review the relevant literature on this. As an example, you may check to the following studies: https://doi.org/10.3390/children10060950
https://doi.org/10.1007/s00784-021-03792-0
Comments on the Quality of English LanguageA small/minor English revision is needed.
Author Response
Comments:
The sample size is not clear. Please clarify it more explicitly in order to eliminate the ambiguity. To do this, refer to a previous similar study.
In the revision comment, when it was requested to specify how reliability is measured in repeated measurements, it was not meant to refer to the name and accuracy of the measuring device. Reliability in repeated measurements means the following: you take an initial measurement, and to test whether this measurement is reliable, you perform a remeasurement on 20-25% of the total sample randomly selected after a period of time (for example, after 2-4 weeks). Then, you demonstrate the reliability of the measurements by presenting the intra-class correlation coefficients (ICCs) for the two measurements. Please review the relevant literature on this. As an example, you may check to the following studies: https://doi.org/10.3390/children10060950
https://doi.org/10.1007/s00784-021-03792-0
Response:
Thanks for the time and effort! We appreciate your comments that can do nothing but improve the quality of our research outcomes.
Comments
- We conducted this study in the third quarter of 2024. When we completed the study, we transferred the anatomical material (human heads) for cremation based on the internal regulations of the morgue of the scientific unit. Before that, for further research, we removed the mandibles from the heads to store them in a dry form. We can not implement test-retest reliability (retaking measurements and indices recalculation).
- Our paper underwent grammar and spell check.